# NB-LRR Lineage-Specific Equipment Is Sorted Out by Sequence Pattern Adaptation and Domain Segment Shuffling

**DOI:** 10.3390/ijms232214269

**Published:** 2022-11-17

**Authors:** Giuseppe Andolfo, Antimo Di Donato, Maria Raffaella Ercolano

**Affiliations:** Department of Agricultural Sciences, University of Naples “Federico II”, Via Università 100, Portici, 80055 Naples, Italy

**Keywords:** plant genome, *R*-gene, immunity system, protein domain, motif architecture, comparative analysis

## Abstract

The nucleotide-binding and leucine-rich repeat (NB-LRR) genes, also known as resistance (*R*)-genes, play an important role in the activation of immune responses. In recent years, large-scale studies have been performed to highlight the diversification of plant NB-LRR repertories. It is well known that, to provide new functionalities, NB-LRR sequences are subject to duplication, domain fusions and acquisition and other kinds of mutations. Although some mechanisms that govern NB-LRR protein domain adaptations have been uncovered, to retrace the plant-lineage-specific evolution routes of R protein structure, a multi-genome comparative analysis was performed. This study allowed us to define groups of genes sharing homology relationships across different species. It is worth noting that the most populated groups contained well-characterized R proteins. The arsenal profile of such groups was investigated in five botanical families, including important crop species, to underline specific adaptation signatures. In addition, the dissection of 70 NB domains of well-characterized *R*-genes revealed the NB core motifs from which the three main R protein classes have been diversified. The structural remodeling of domain segments shaped the specific NB-LRR repertoires observed in each plant species. This analysis provided new evolutionary and functional insights on NB protein domain shuffling. Taken together, such findings improved our understanding of the molecular adaptive selection mechanisms occurring at plant R loci.

## 1. Introduction

Plants, to defend themselves against pathogens, have developed a multilevel protective and surveillance network of pathogen receptor genes [1,2]. The innate immune system of plants is traditionally described as composed of two recognition layers of pathogenic invasion, named pathogen-associated molecular-pattern-triggered immunity (PTI) and effector-triggered immunity (ETI) [3], which cooperate mutually [4,5]. ETI responses are primed by specialized receptors in the plant called resistance (*R*-) genes, also known as nucleotide-binding–leucine-rich repeat (NB-LRR) genes [6,7].

Following complex evolution trajectories, the first NB-LRR domain associations date back to over 3.5 billion years ago [8,9]. Recently, it was proposed that convergent evolution, via horizontal gene transfer, may have generated the NB-LRR supra-domain structure in separate Chlorophyta and Streptophyta lineages [9]. Furthermore, a large diversification occurred in early plants thanks to the recombination of existing units or the establishment of novel combinations.

NB-LRR genes can be classified into three major protein classes, based on different N-terminus architecture, namely TIR-NB-LRR (TNL), CC-NB-LRR (CNL) and RPW8-NB-LRR (RNL). TNLs contain a protein domain with homology to the Drosophila Toll and mammalian Interleukin-1 Receptor (TIR) [10]. CNLs and RNLs present a predicted helical Coiled-Coil (CC) motif [7] and a Resistance to Powdery Mildew 8 (RPW8) domain, respectively [11,12]. The evolution of NB-LRR has been linked to tandem duplications occurring in specific *R*-gene clustering regions [13]. Indeed, gene duplication is an important source for generating new genetic material for inducing biological innovation. Host plants employ diverse families of NB-LRR genes, resulting from gene duplication events and further structural diversification of individual protein domains [14]. To date, much has been done to decipher the structural and functional reassembling of NB-LRR protein domains [8,9,15,16], but it remains unclear how NB-LRR families have been diversified and scattered across diverging plant lineages.

The domain architecture of *R*-genes is consistent with a role in pathogen recognition and defense response signaling. The N-terminal CC or TIR domains are typically described as required for downstream signaling following the perception of pathogens [17,18]. The extremely variable C-terminal LRR domain provides target specificity. The highly conserved NB domain regulates the protein ON/OFF state by binding and hydrolyzing ADP and GTP [19], and contains highly conserved motifs involved in intra- and extra-molecular interactions [20,21]. These include the motifs hhGRExE, P-loop (Walker A/kinase 1), RNBS-A, kinase 2 (Walker B), kinase 3a, RNBS-B, RNBS-C, GLPL and RNBS-D and MHD [22,23]. The high level of conservation of this amino acid region makes the NB domain very useful for studying the genomic architecture of NB-LRR gene family [24]. Recently, the evolutionary diversification of RNL protein class has been unraveled from NB motif combinations along land plant lineages [24]. However, the NB domain diversification analysis at the subdomain level has not yet been fully addressed.

In this study, we performed comparative genomic analyses of NB-LRR genes annotated in 104 proteomes evaluating large-scale orthology relationships. A focus on *R*-gene homolog profiles of five crop families was conducted to highlight lineage-specific evolution routes. The diversification of *R*-gene classes was further investigated by the means of NB domain shuffling. Indeed, the analysis of well-characterized R proteins was used to infer NB lineage-specific motif structure. Our findings provide novel evolutionary insights into the diversification of NB-LRR immune receptors in plants.

## 2. Results

### 2.1. Diversification of NB-LRR Gene Receptors during Green Plant Evolution

To retrace the key steps which have marked the evolution of plant *R*-genes, we explored a total of 34,979 sequences that encode domains similar to plant R proteins (Appendix A). These NB-LRR genes were annotated in 104 out of 120 analyzed genomes, representing over 50 taxa (Appendix A). The evolutionary path from ancestral R protein domains to the supra domain (NB-LRR) required approx. 3.5 billion years (Figure 1). The first NB-LRR assembly was retrieved in *Chromochloris zofingiensis* and subsequent R-domain-reassembling events were observed in nonflowering plants [9]. In the plant kingdom, NB-LRR gene family size exhibited a considerable variability (from 3.2% in *Coffea canephora* to 0.04% in *Klebsormidium flaccidum* and *Chromochloris zofingiensis*) (Appendix A). NB-LRR paralogs accounted for up to ~95% of the total complement in *Lactuca sativa* and *Nicotiana tabacum* (Appendix A). On the basis of the positive correlation (Pearson’s r = 0.76) between the number of NB-LRR gene clusters and NB-LRR paralogs (Appendix A), it may be assumed that they play a leading role in tandem duplications in the *R*-gene family expansion.

### 2.2. Lineage-Specific *R*-gene Profiles in Major Crops

Orthology inference analysis allowed to identify 1675 NB-LRR orthogroups. Approximately 36% (12,774) of the total analyzed proteins were grouped in 41 orthogroups, containing 70 functionally characterized R proteins (Appendix A). Solanaceae and Poaceae showed a conspicuous number of orthogroups and paralogs. Instead, Brassicaceae and Cucurbitaceae diversified their NB-LRR complement starting from a limited number of initial sequences. A potential increment of NB-LRR neo-functionalization events was suggested by correlation (Pearson’s r = 0.82) between the number of orthogroups and the size of the NB-LRR gene family (Table 1 and Appendix A). Intriguingly, the largest orthologues group, missing Fabaceae, Solanaceae and Rosaceae family members, included several NB-LRR duplications in Poaceae (Figure 2). By contrast, the second most populated group, containing four solanaceous TNLs (*Gro1.4*, *N*, *RY-1* and *Bs4*), lacked Poaceae homologs and it was highly represented in the other important crop families (Appendix A). The *Rpi-blb1* group was present in all crops except for Brassicaceae and it was highly duplicated in the legume and the nightshade families (Appendix A). The *ADR1* group was conserved in all crop families, and its homologs were found also in early land plants. Moreover, the characteristic Resistance to Powdery Mildew 8 (RPW8) domain of *ADR1* was detected in several genes belonging to its orthogroup (Appendix A). By contrast, *NRG1* copies were conserved in 48 analyzed eudicot genomes belonging to the analyzed plant family, with a number of genes ranging from 1 to 17 (Appendix A). *Fom-2* orthologs were found in Rosaceae (62) and Solanaceae (81) families. Interestingly, out of a total of 587 *Fom-2* homologs, 194 were detected in coffee genome (Appendix A). Our analysis identified *NRC* homologs in all analyzed Superasterid genomes and underlined a high conservation in the nightshades (Appendix A). Grasses and nightshades possessed several highly duplicated private groups, containing cloned genes conferring resistance to fungi and bacteria, respectively.

### 2.3. NB Domain Diversification in a *R*-genes Core Collection

To outline the evolutionary routes emerging from the NB domain diversification, a motif-based sequences analysis was carried out in 70 functionally characterized *R*-genes (Figure 3). The maximum likelihood analysis displayed a clear distinction between TNL, RNL and CNL (CNL-1 to CNL-5) gene classes (Figure 3A). The *R*-genes collapsed into seven clades that have high sequence similarities and were supported by bootstrap values  > 50%.

The NB Pfam domains of the well-characterized *R*-genes were divided into 30 ungapped motifs (Appendix A). The clade-specific motif structures are shown in Figure 3B and are marked with reference to clade-specific colors (Figure 3A). A total of 13 motifs were conserved in all analyzed *R*-genes (black in Figure 3B). Most likely, the evolution events characterizing the diversification of *R*-gene classes in plants originated from a limited core of motifs. The TNL and CNL classes showed 8 and 10 specific motifs, respectively (blue and violet in Figure 3B). In addition, the clade CNL1, CNL2, CNL3, CNL4 and CNL5, were univocally characterized by motifs 1, 2, 4, 1 and 9, respectively. Hierarchical cluster analysis allowed us to reveal the dynamics leading to the birth–death of specific NB domain motifs during R protein diversification (Figure 3C). The heatmap dendrogram clustered *R*-genes on the basis of presence/absence of common NB motifs, independently from their physical order in the NB domain sequence (Figure 3C). Eight different groups were identified and marked (red triangles in Figure 3C).

Group I included the RNLs (*NRG1* and *ADR1*), the TNLs (*Rps4* and *P2*), and the CNLs (*Dm3* and *VAT*). *NRG*, cloned in *Nicotiana benthamiana*, had a large diversification in 227 orthologs belonging to 44 species (Table 2). *ADR1* and *Rps4* were first identified in *Arabidopsis thaliana* and then retrieved in 77 and 10 genomes, respectively. Instead, Dm3 and *P2* copies were found only in *Lactuca sativa* and *Linum usitatissimum*, respectively. Finally, *VAT* gene, cloned in *Cucumis melo*, showed a limited diversification in six genomes. These six genes, located in the boundary zone of TNL, RNL and CNL gene classes, have subsequently embarked on a different evolutionary route (Figure 3A).

The pattern of group I included the basic motif combination from which, through little changes, arose the NB domain architecture of *R*-gene classes. Interestingly, a total of 643 orthologs of functionally characterized *R*-genes belonging to group I were found (Table 2).

Groups II, V and VII included the genes of clade CNL5 (Figure 3B). The remaining TNL genes collapsed within groups III and IV; the latter differed from the previous for the presence of the motif M28a (Figure 3B). Ten CNL genes of CNL2 and CNL3 clades clustered into group VI. Finally, group VIII included 27 CNL genes belonging to five different phylogenetic clades (Figure 3).

## 3. Discussion

To fight a multitude of phytopathogens, plants have diversified a wide defense arsenal from a successful supra-domain assembly originating 3.5 billion years ago [8,15]. About 35,000 NB-LRR genes were identified in 104 genomes using a protein domain search approach [8,23]. To minimize the risk of bias in *R*-gene identification, we used gene sets from the soft-masked versions of the genome assemblies [9,25]. However, small imprecision could still be present in our NB-LRR annotation [9]. Automated gene predictions could lead to incomplete representation of *R*-genes within gene sets [26,27]. In all species, including non-vascular land plants, a conspicuous number of NB-LRR paralogs, varying in order of magnitude across plant species, was identified [28,29,30]. A NB-LRR burst expansion and subsequent adjustments of gene structure were observed in early land plants [9]. The species’ lifestyle and the selection pressures derived from pathogen co-evolution allowed the establishment of lineage-specific NB-LRR repertories [9]. The strong correlation found between the number of NB-LRR gene clusters and NB-LRR paralogs underlined that cluster organization promoted gene diversification. Tandem duplications, unequal crossing-over and transposition events were able to maintain a diverse array of genes to retain advantageous resistance specificities [31]. NB-LRR adaptation is based on molecular mechanisms and evolutionary forces not completely unveiled. Recent investigations on Triticaceae genomes revealed that an increased dosage or sub-/neo-functionalization in agronomically important genes occurred [32]. Tandem duplications and the activity of transposable elements could have a main role in generating NB-LRR copies with new characteristics [32].

Genome-wide analysis of NB-LRR families relationships provided insights into their evolutionary history. The complex domains arrangement of NB-LRR genes and the wide spectrum of mutations reflect the need of adjustments driven by the dynamic “arms race” among *R*-genes and pathogens in the different species. It is worth to know that the 50% of NB-LRR gene copies retrieved in our analysis belong to few orthogroups, including well-characterized R proteins, and the rest are spread across more than 1500 different group variants. Following duplication, genes can accumulate mutations that can be retained, if advantageous, or lost in the span of a few million years, if deleterious [32]. Among duplicate genes, new functions are expected to emerge when a new adaption to environment is required. Duplication of specific genes resulted in divergent evolution among botanical families. Looking to the orthologous *R*-gene profiles of main crop families, it results clear that Poaceae highly duplicated the MLA members, absent in other botanical families. By contrast, the TNL group including four Solanaceus genes (Gro1.4, N, RY-1, Bs4) lacks Poaceae homologs and it is highly represented in the other important crop families.

The emergence of a variable number of specific duplicated genes drove the species-specific divergence from an initial core set of limited sequences. Recent duplicates with highly sequence-similarity are expected to be located within specific regions of the genome. Indeed, NB-LRR groups have expanded in each genome, due to duplication events occurred in specific loci [25]. The profile of a given species was shaped by fixing useful duplicated sequence and removing harmful variants.

These findings provide important foundational knowledge for understanding NLR evolution and empowering plant disease resistance. Furthermore, extensive functional studies have shown that the different domains have to be finely matched for optimal specificity and robustness of NB-LRR signaling [33].

The characteristic motifs of the NB domain have been extensively employed to distinguish the different R protein classes [34] and to define resistance gene homologs in model and crop species [35,36,37]. The NB domain is involved in the controls of protein functioning [38], the binding to the nucleotide ATP enables an active conformation while the binding to ADP determines an inactive conformation [39].

The activity and specificity of NB-LRR variants can be drastically altered by its segment sequence changes [40,41]. The role of specific NB motifs have been functionally disclosed: the Walker A motif (or P-loop) is important for nucleotide binding, Walker B motif is required for ATP hydrolysis, the conserved “GLPL” (glycine-leucine-proline-leucine) and the “MHD” (methionine-histidine-aspartate) motifs, when mutated, usually results in an autoactive phenotype [40,42]. Recently the analyses of segment motifs within NB domains allowed to rebuild the *R*-gene evolution [24]. Interestingly, the ancestral pattern of motifs was shared among NB-LRR genes belonging to the three major R protein classes (CNL, TNL and RNL). Most likely, the NB domain segment diversification together with N and C-terminal regions shuffling has contributed to functional specialization of NB-LRR protein classes [9,43]. NB motif evolution is less striking than the introgression or loss of a protein domain, but not less important as evidenced by the ADR1 and NRG1 lineage evolution [24].

## 4. Materials and Methods

### 4.1. Taxa Dataset and NB-LRR Gene Annotation

The genomic data of 120 organisms were retrieved from Phytozome (http://phytozome.jgi.doe.gov, accessed on 13 November 2022) and other plant genome websites (Appendix A). The proteomes of our taxa data set were initially scanned for the Hidden Markov Model (HMM) profiles of Nucleotide-Binding (Pfam PF00931) and Leucine-Rich Repeat domains (Panther PTHR11017:SF191) in HMMER v3 using “hmmsearch” with an expected value (e-value) threshold of <1 × 10^2^. Furthermore, additional NB-LRR candidates were identified by mapping *R*-gene motifs, released by Andolfo et al., [40], to the proteome data set using BlastP (E-value 1 × 10^2^). The domain architecture of protein sequences identified by HMMER and BLAST was further confirmed using the programs Pfam, Panther, SuperFamily and CDD as implemented in the InterProScan v5 software with default parameters [41]. The information archived in APG IV (Angiosperm Phylogeny Group) [42], Angiosperm Phylogeny Website (http://www.mobot.org/MOBOT/research/APweb/welcome.html, accessed on 13 November 2022) and “The Tree of Life Web Project” (http://tolweb.org/tree/, accessed on 13 November 2022)) were used to generate a dendrogram of analyzed species (Appendix A).

### 4.2. Identification of Orthologous Groups and Physical R-Clusters

A subset of functionally characterized *R*-genes was used for a reciprocal best hit analysis (threshold E-value < 1 × 10^−5^) (Appendix A). The orthologuos groups were obtained using OrthoMCL tool [44] with default parameters. The association of reference *R*-genes (http://prgdb.crg.eu/, accessed on 13 November 2022)) and relative orthogroup was detected using Best Hit method (BlastP, E-value < 1 × 10^−5^) (Appendix A).

Physical clustering of candidate genes was detected using a customized script with GFF file of corresponding genomes and Bedtools [45]. If two NB-LRR genes were separated by no more than eight other genes, they were considered to be located at the same gene cluster [46].

### 4.3. Maximum Likelihood Analysis

Evolutionary analyses were conducted using MEGA7 [47]. The phylogenetic relationships of R proteins were inferred using the maximum likelihood method based on Jones et al. [6] w/freq. model. The model with the lowest Bayesian information criterion score was considered to better describe the substitution pattern. The bootstrap consensus tree inferred from 100 replicates was taken to represent the evolutionary history of the sequences analyzed [48]. The trees were drawn to scale, with branch lengths measured by estimating the number of substitutions per site.

### 4.4. De Novo Prediction of NB-Encoding Genes Motifs

The Multiple EM for Motif Elicitation (MEME) (http://meme-suite.org/, accessed on 13 November 2022)) algorithm [49] was used to decompose in motifs (Appendix A) the NB Pfam domain (PF00931) of NB-LRR protein dataset [24]. The motifs were enumerated from M1 to M30, and when the same motif (e.g., M6) was identified more than once in a NB domain sequence the motif ID was further specified by a letter (e.g., M6a and M6b). The analysis was carried out using the default cut-off value for statistical confidence. The Motif Alignment and Search Tool (MAST) (http://meme-suite.org/, accessed on 13 November 2022)) [49] was also used to confirm the presence of MEME motifs previously identified (Appendix A), using the default setting. A heat map was generated starting from a motif–presence matrix using the ‘GPLOTS’ R software package [50].

## Figures and Tables

**Figure 1 ijms-23-14269-f001:**
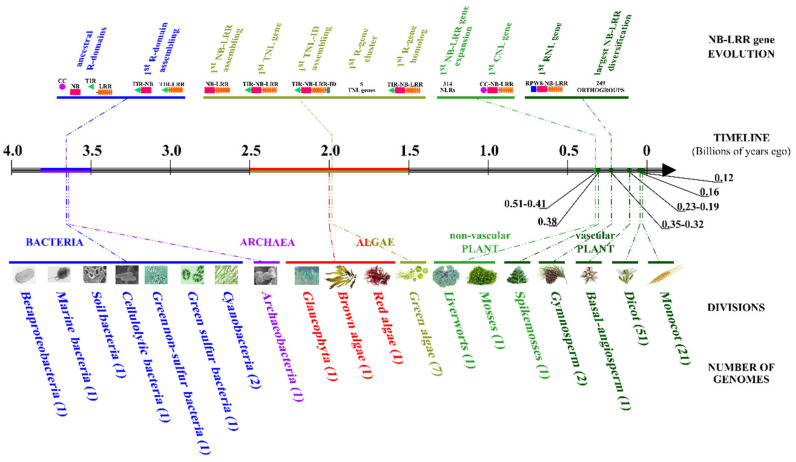
Schematic representation of crucial events which gave rise to NB-LRR gene family in green plants. At the top, the key points of *R*-gene evolution are connected by dotted lines to timeline and relative taxonomic group. Ancestral R protein domains, first R domain assembling, first NB-LRR association, first TNL gene, first *R*-gene cluster, first NB-LRR gene expansion, first CNL gene, first *R*-gene homolog, first RNL association and largest NB-LRR diversification are reported. Finally, the division to which belong the genomes is indicated by pictures.

**Figure 2 ijms-23-14269-f002:**
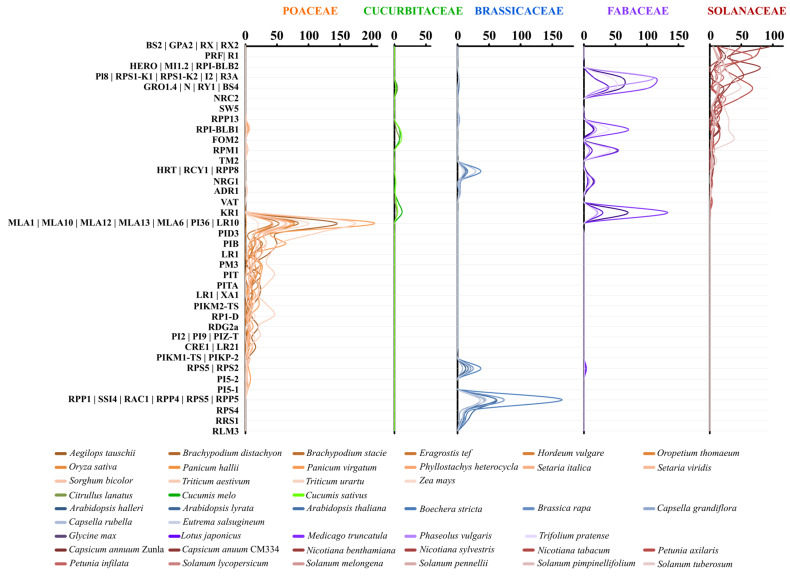
*R*-gene family profiles in over 40 flowering plant species (16 Poaceae in orange, 3 Cucurbitaceae in green, 8 Brassicaceae in blue, 5 Fabaceae in violet and 12 Solanaceae in red). The number of *R*-gene homologs for each analyzed crop are reported in the upper part of graphic; the *R*-gene families are listed on the left.

**Figure 3 ijms-23-14269-f003:**
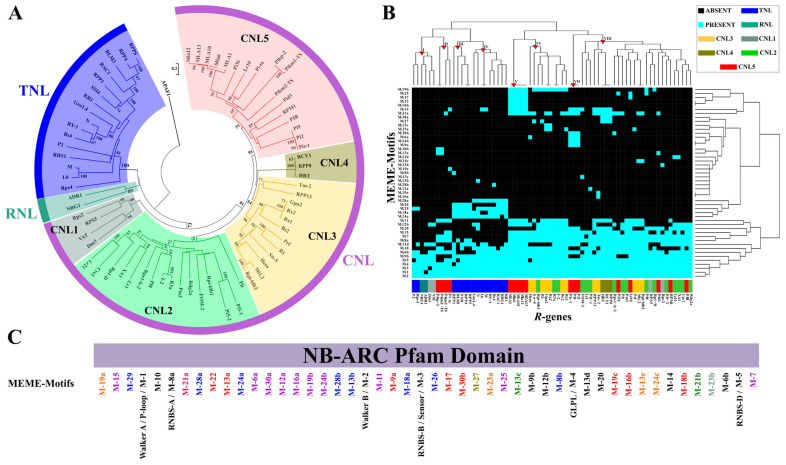
Diversification patterns of the NB domain in the plant *R*-gene family. (**A**) Phylogenetic tree of 70 well-characterized *R*-genes. The seven phylogenetic clades are indicated with different colors. (**B**) Organization of motifs along the NB Pfam sequence. Motif IDs (M1-M30) and physical order are indicated. The colors are related to panel A. (**C**) Hierarchical cluster analysis of NB motifs. The color of each cell of the heatmap is based on the presence/absence of a specific NB motif in that *R*-gene. Eight *R*-gene clusters are indicated using red triangles. The colors of *R*-genes are related to panel A.

**Table 1 ijms-23-14269-t001:** Number of orthogroups, paralogs and gene clusters in five principal crop families.

Plant Family	Average Number of Orthogroups	Average Number of Paralogs	Average Number of Gene Clusters
Brassicaceae	52 (39–72)	171 (105–322)	128 (119–135)
Fabaceae	70 (49–89)	470 (295–803)	392 (220–706)
Solanaceae	96 (53–156)	306 (157–645)	338 (141–622)
Poaceae	94.5 (41–149)	384 (25–1033)	172 (15–666)
Cucurbitaceae	24 (20–27)	54 (41–62)	29 (-)

The average number refers to the arithmetic mean. In brackets are reported the numeric range (minimum and maximum) of orthogroups, paralogs and clusters.

**Table 2 ijms-23-14269-t002:** Orthologous genes to *R*-genes group I reported in Figure 3.

*R*-Gene of Group I	Protein Class	Orthogroup ID	Number of Orthologs	Number of Genomes
*NRG1*	RNL	OG1028	227	44
*ADR1*	RNL	OG1033	193	77
*VAT*	CNL	OG1169	17	6
*RPS4*	TNL	OG1043	128	10
*Dm3*	CNL	OG1093	44	1
*P2*	TNL	OG1106	34	1

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
