# Peer review of "NB-LRR Lineage-Specific Equipment Is Sorted Out by Sequence Pattern Adaptation and Domain Segment Shuffling"

_ijms, 2022, doi:10.3390/ijms232214269_

Round 1
Reviewer 1 Report
In this manuscript, authors investigate the functional domains of nucleotide-binding and leucine-rich repeat (NLR) genes across multiple plant species. Overall, this study is well designed and the manuscript is well organized. The methods and analyses can support the conclusions. This study will improve our understanding of the mechanism of plant NLR genes. It will benefit the manuscript if some minor concerns below can be addressed.
(1) Materials and Methods section: Add detailed running parameters of the bioinformatics tools used in this study, for example, HMMER, BLAST et al. This will help the people to repeat this study in the future.
(2) In Figure 1, the picture/carton of each species should be cited if it was not drawn by the authors.
(3) In Table 1, what is the average, mean or median? Please clarify in the note.
(4) In Discussion section, the shortcomings of this study should be also discussed.
Author Response
Dear Editor,
enclosed please find the revised version of manuscript ijms-1985667 "NB-LRR lineage-specific equipment is sorted out by sequence pattern adaptation and domain segment shuffling" by Andolfo et al. During revision, we have done our best to address the valuable comments of the reviewers. Below, we provide a point-to-point reply to each comment. Changes with respect to the previous version are highlighted in a separate manuscript file in red.
Overall, we believe that the revised manuscript fully addresses all the comments and we are looking forward to seeing our work published in the International Journal of Molecular Science.
Best regards,
Maria Raffaella Ercolano on behalf of all the co-authors
Comments and Suggestions for Authors
In this manuscript, authors investigate the functional domains of nucleotide-binding and leucine-rich repeat (NLR) genes across multiple plant species. Overall, this study is well designed and the manuscript is well organized. The methods and analyses can support the conclusions. This study will improve our understanding of the mechanism of plant NLR genes. It will benefit the manuscript if some minor concerns below can be addressed.
- We appreciate the time Reviewer#1 has spent in reading and revising this manuscript and we believe that their suggestions helped to improve our manuscript. All changes in the revised version are highlighted in the text. Please, find our responses to each of your suggestions/comments below.
- Materials and Methods section: Add detailed running parameters of the bioinformatics tools used in this study, for example, HMMER, BLAST et al. This will help the people to repeat this study in the future.
- We thank Reviewer#1 for pointing out this important detail. We have added more information on the setting parameters of the used bioinformatic tools.
- In Figure 1, the picture/carton of each species should be cited if it was not drawn by the authors.
- The pictures used in Figure 1 are free copyright and modified by authors.
- In Table 1, what is the average, mean or median? Please clarify in the note.
- We indicated it.
- In Discussion section, the shortcomings of this study should be also discussed.
- We have added a section (183-187 lines) on “shortcomings of this study” in the Discussion paragraph.
Reviewer 2 Report
The work represents a very broad contribution to the knowledge of NB-LRR genes in terms of the characteristics of their domains and the evolutionary process that allowed these sequences to be amplified while conserving their resistance function.
The characterization of the different domains allows them to postulate a mechanism of random duplication to achieve the expansion of the family of R genes.
Considering the number of orthologous genes groups that code for functional R proteins and the size of the NB-LRR gene families, they postulate a potential increase in events that lead to new functions of NB-LRR genes.
The characterization of the NB domain generates very valuable information to identify new resistance genes and expand the possibilities of selecting new resistant crops.
Author Response
Comments and Suggestions for Authors
The work represents a very broad contribution to the knowledge of NB-LRR genes in terms of the characteristics of their domains and the evolutionary process that allowed these sequences to be amplified while conserving their resistance function. The characterization of the different domains allows them to postulate a mechanism of random duplication to achieve the expansion of the family of R genes. Considering the number of orthologous genes groups that code for functional R proteins and the size of the NB-LRR gene families, they postulate a potential increase in events that lead to new functions of NB-LRR genes. The characterization of the NB domain generates very valuable information to identify new resistance genes and expand the possibilities of selecting new resistant crops.
- We thank Reviewer#2 for the positive response to our work and we are glad that our effort was appreciated.